# *Drop-Bottleneck*: Learning Discrete Compressed Representation for Noise-Robust Exploration

**Jaekyeom Kim, Minjung Kim, Dongyeon Woo & Gunhee Kim**
Department of Computer Science and Engineering
Seoul National University, Seoul, Republic of Korea
`jaekyeom@snu.ac.kr,minjung.kim@vl.snu.ac.kr,{woody0325,gunhee}@snu.ac.kr`
`http://vision.snu.ac.kr/projects/db`

## Abstract

We propose a novel information bottleneck (IB) method named *Drop-Bottleneck*, which discretely drops features that are irrelevant to the target variable. Drop-Bottleneck not only enjoys a simple and tractable compression objective but also additionally provides a deterministic compressed representation of the input variable, which is useful for inference tasks that require consistent representation. Moreover, it can jointly learn a feature extractor and select features considering each feature dimension's relevance to the target task, which is unattainable by most neural network-based IB methods. We propose an exploration method based on Drop-Bottleneck for reinforcement learning tasks. In a multitude of noisy and reward sparse maze navigation tasks in VizDoom (Kempka et al., 2016) and DM-Lab (Beattie et al., 2016), our exploration method achieves state-of-the-art performance. As a new IB framework, we demonstrate that Drop-Bottleneck outperforms Variational Information Bottleneck (VIB) (Alemi et al., 2017) in multiple aspects including adversarial robustness and dimensionality reduction.

## 1 Introduction

Data with noise or task-irrelevant information easily harm the training of a model; for instance, the noisy-TV problem (Burda et al., 2019a) is one of well-known such phenomena in reinforcement learning. If observations from the environment are modified to contain a TV screen, which changes its channel randomly based on the agent's actions, the performance of curiosity-based exploration methods dramatically degrades (Burda et al., 2019a;b; Kim et al., 2019; Savinov et al., 2019).

The information bottleneck (IB) theory (Tishby et al., 2000; Tishby & Zaslavsky, 2015) provides a framework for dealing with such task-irrelevant information, and has been actively adopted to exploration in reinforcement learning (Kim et al., 2019; Igl et al., 2019). For an input variable $X$ and a target variable $Y$, the IB theory introduces another variable $Z$, which is a compressed representation of $X$. The IB objective trains $Z$ to contain less information about $X$ but more information about $Y$ as possible, where the two are quantified by mutual information terms of $I(Z; X)$ and $I(Z; Y)$, respectively. IB methods such as Variational Information Bottleneck (VIB) (Alemi et al., 2017; Chalk et al., 2016) and Information Dropout (Achille & Soatto, 2018) show that the compression of the input variable $X$ can be done by neural networks.

In this work, we propose a novel information bottleneck method named *Drop-Bottleneck* that compresses the input variable by discretely dropping a subset of its input features that are irrelevant to the target variable. Drop-Bottleneck provides some nice properties as follows:

- The compression term of Drop-Bottleneck's objective is simple and is optimized as a tractable solution.

- Drop-Bottleneck provides a *deterministic* compressed representation that still maintains majority of the learned indistinguishability *i.e.* compression. It is useful for inference tasks that require the input representation to be consistent and stable.

- Drop-Bottleneck jointly trains a feature extractor and performs feature selection, as it learns the feature-wise drop probability taking into account each feature dimension's relevance to the target task. Hence, unlike the compression provided by most neural network-based IB

      methods, our deterministic representation *reduces* the feature dimensionality, which makes the following inference better efficient with less amount of data.

- Compared to VIB, both of Drop-Bottleneck's original (stochastic) and deterministic compressed representations can greatly improve the robustness to adversarial examples.

Based on the newly proposed Drop-Bottleneck, we design an exploration method that is robust against noisy observations in very sparse reward environments for reinforcement learning. Our exploration maintains an episodic memory and generates intrinsic rewards based on the predictability of new observations from the compressed representations of the ones in the memory. As a result, our method achieves state-of-the-art performance on multiple environments of VizDoom (Kempka et al., 2016) and DMLab (Beattie et al., 2016). We also show that combining our exploration method with VIB instead of Drop-Bottleneck degrades the performance by meaningful margins.

Additionally, we empirically compare with VIB to show Drop-Bottleneck's superior robustness to adversarial examples and ability to reduce feature dimensionality for inference with ImageNet dataset (Russakovsky et al., 2015). We also demonstrate that Drop-Bottleneck's deterministic representation can be a reasonable replacement for its original representation in terms of the learned indistinguishability, with Occluded CIFAR dataset (Achille & Soatto, 2018).

## 2    Related Work

### 2.1    Information Bottleneck Methods

There have been a number of IB methods that are approximations or special forms of the original IB objective. Variational Information Bottleneck (VIB) (Alemi et al., 2017) approximates the original IB objective by establishing variational bounds on the compression and prediction terms. Chalk et al. (2016) propose the same variational bound on the IB objective in the context of sparse coding tasks. Conditional Entropy Bottleneck (CEB) and Variational Conditional Entropy Bottleneck (VCEB) (Fischer, 2020; Fischer & Alemi, 2020) use an alternative form of the original IB objective derived under the Minimum Necessary Information (MNI) criterion to preserve only a necessary amount of information. The IB theory (Tishby et al., 2000) has been used for various problems that require restriction of information or dealing with task-irrelevant information. Information Dropout (Achille & Soatto, 2018) relates the IB principle to multiple practices in deep learning, including Dropout, disentanglement and variational autoencoding. Moyer et al. (2018) obtain representations invariant to specific factors under the variational autoencoder (VAE) (Kingma & Welling, 2013) and VIB frameworks. Amjad & Geiger (2019) discuss the use of IB theory for classification tasks from a theoretical point of view. Dai et al. (2018) employ IB theory for compressing neural networks by pruning neurons in networks. Schulz et al. (2020) propose an attribution method that determines each input feature's importance by enforcing compression of the input variable via the IB framework.

Similar to our goal, some previous research has proposed variants of the original IB objective. Deterministic information bottleneck (DIB) (Strouse & Schwab, 2017) replaces the compression term with an entropy term and solves the new objective using a deterministic encoder. Nonlinear information bottleneck (NIB) (Kolchinsky et al., 2019) modifies the IB objective by squaring the compression term and uses a non-parametric upper bound on the compression term. While DIB is always in the deterministic form, we can flexibly choose the stochastic one for training and the deterministic one for test. Compared to NIB, which is more computationally demanding than VIB due to its non-parametric upper bound, our method is faster.

### 2.2    Reinforcement Learning with Information Bottleneck Methods

The IB theory has been applied to several reinforcement learning (RL) tasks. Variational discriminator bottleneck (Peng et al., 2019) regulates the discriminator's accuracy using the IB objective to improve adversarial training, and use it for imitation learning. Information Bottleneck Actor Critic (Igl et al., 2019) employs VIB to make the features generalize better and encourage the compression of states as input to the actor-critic algorithm. Curiosity-Bottleneck (Kim et al., 2019) employs the VIB framework to train a compressor that compresses the representation of states, which is still informative about the value function, and uses the compressiveness as exploration signals. InfoBot (Goyal et al., 2019) proposes a conditional version of VIB to improve the transferability of a goal-conditioned policy by minimizing the policy's dependence on the goal. Variational bandwidth

bottleneck (Goyal et al., 2020) uses a modified, conditional version of VIB, and solves RL tasks with privileged inputs (*i.e.* valuable information that comes with a cost). Our exploration method differs from these methods in two aspects. First, we propose a new information bottleneck method that is not limited to exploration in RL but generally applicable to the problems for which the IB theory is used. Second, our method generates exploration signals based on *the noise-robust predictability i.e.* the predictability between noise-robust representations of observations.

## 3 DROP-BOTTLENECK

### 3.1 PRELIMINARIES OF INFORMATION BOTTLENECK

Given an input random variable $X$, the information bottleneck (IB) framework (Tishby et al., 2000; Tishby & Zaslavsky, 2015) formalizes a problem of obtaining $X$'s compressed representation $Z$, which still and only preserves information relevant to the target variable $Y$. Its objective function is

$$\text{minimize} -I(Z;Y) + \beta I(Z;X), \tag{1}$$

where $\beta$ is a Lagrangian multiplier. The first and second terms are prediction and compression terms, respectively. The prediction term $I(Z;Y)$ encourages $Z$ to preserve task-relevant information while the compression term $I(Z;X)$ compresses the input information as much as possible.

As reviewed in the previous section, there have been several IB methods (Alemi et al., 2017; Chalk et al., 2016; Achille & Soatto, 2018; Strouse & Schwab, 2017; Kolchinsky et al., 2019), among which the ones derived using variational inference such as Variational Information Bottleneck (VIB) (Alemi et al., 2017) have become dominant due to its applicability to general problems.

### 3.2 DROP-BOTTLENECK

We propose a novel information bottleneck method called *Drop-Bottleneck* (DB), where the input information is compressed by *discretely* dropping a subset of input features. Thus, its compression objective is simple and easy to optimize. Moreover, its representation is easily convertible to a deterministic version for inference tasks (Section 3.3), and it allows joint training with a feature extractor (Section 3.4). While discrete dropping of features has been explored by prior works including Dropout (Srivastava et al., 2014), DB differs in that its goal is to assign different drop probabilities to feature variables based on their relevance to the target variable.

For an input variable $X = [X_1, \ldots, X_d]$ and a drop probability $\boldsymbol{p} = [p_1, \ldots, p_d] \in [0, 1]^d$, we define its compressed representation as $Z = C_{\boldsymbol{p}}(X) = [c(X_1, p_1), \ldots, c(X_d, p_d)]$ such that

$$c(X_i, p_i) = b \cdot \text{Bernoulli}(1 - p_i) \cdot X_i, \quad \text{where} \quad b = \frac{d}{d - \sum_k p_k}, \tag{2}$$

for $i = 1, \ldots, d$. That is, the compression procedure *drops* the $i$-th input feature (*i.e.* replaced by zero) with probability $p_i$. Since the drop probability is to be learned, we use a scaling factor $b$ to keep the scale of $Z$ constant. We use a single scaling factor for all feature dimensions in order to preserve the relative scales between the features.

With the definition in Equation (2), we derive the compression term of DB to minimize as

$$I(Z;X) = \sum_{i=1}^{d} I(Z_i; X_1, \ldots, X_d | Z_1, \ldots, Z_{i-1}) \tag{3}$$

$$= \sum_{i=1}^{d} [I(Z_i; X_i | Z_1, \ldots, Z_{i-1}) + I(Z_i; X_1, \ldots, X_d \setminus X_i | Z_1, \ldots, Z_{i-1}, X_i)] \tag{4}$$

$$= \sum_{i=1}^{d} I(Z_i; X_i | Z_1, \ldots, Z_{i-1}) \leq \sum_{i=1}^{d} I(Z_i; X_i) = \hat{I}(Z;X) \tag{5}$$

using that $Z_i \perp\!\!\!\perp X_1, \ldots, X_{i-1}, X_{i+1}, \ldots, X_d | Z_1, \ldots, Z_{i-1}, X_i$ and $Z_i \perp\!\!\!\perp Z_1, \ldots, Z_{i-1} | X_i$. Note that $\hat{I}(Z;X) - I(Z;X) = \left( \sum_{i=1}^{d} H(Z_i) \right) - H(Z_1, \ldots, Z_d) = TC(Z)$ where $TC(Z)$ is the total correlation of $Z$ and $H(\cdot)$ denotes the entropy, and $\hat{I}(Z;X) = I(Z;X)$ if $X_1, \ldots, X_d$ are independent. To provide a rough view on the gap, due to the joint optimization with the compression

term $\hat{I}(Z;X)$ and the prediction term $I(Z;Y)$, $Z$ becomes likely to preserve less redundant and less correlated features, and $TC(Z)$ could decrease as the optimization progresses.

Finally, DB's new compression term, $\hat{I}(Z;X)$, is computed as

$$\hat{I}(Z;X) = \sum_{i=1}^{d} I(Z_i; X_i) = \sum_{i=1}^{d} \big( H(X_i) - H(X_i|Z_i) \big) \tag{6}$$

$$= \sum_{i=1}^{d} \big( H(X_i) - p_i \cdot H(X_i|Z_i = 0) - (1 - p_i) \cdot H(X_i|Z_i = bX_i) \big) \tag{7}$$

$$\approx \sum_{i=1}^{d} \big( H(X_i) - p_i \cdot H(X_i) - (1 - p_i) \cdot 0 \big) = \sum_{i=1}^{d} H(X_i)(1 - p_i). \tag{8}$$

From Equation (7) to Equation (8), we use the two ideas: (i) $H(X_i|Z_i = 0) = H(X_i)$ because $Z_i = 0$ means it contains no information about $X_i$, and (ii) $H(X_i|Z_i = bX_i) = 0$ because $Z_i = bX_i$ means $Z_i$ preserves the feature (*i.e.* $Z_i$ fully identifies $X_i$) and thus their conditional entropy becomes zero. Importantly, DB's compression term is computed as the simple tractable expression in Equation (8). As the goal of the compression term is to penalize $I(Z;X)$ not $H(X)$, the drop probability $p$ is the only parameter optimized with our compression term. Thus, each $H(X_i)$ can be computed with any entropy estimation method such as the binning-based estimation, which involves quantization for continuous $X_i$, since the computation has no need to be differentiable.

However, one issue of Equation (8) is that $Z$ is not differentiable with respect to $p$ as Bernoulli distributions are not differentiable. We thus use the Concrete relaxation (Maddison et al., 2017; Jang et al., 2016) of the Bernoulli distribution to update $p$ via gradients from $Z$:

$$\text{Bernoulli}(p) \approx \sigma \left( \frac{1}{\lambda} \big( \log p - \log(1 - p) + \log u - \log(1 - u) \big) \right), \tag{9}$$

where $u \sim \text{Uniform}(0, 1)$ and $\lambda$ is a temperature for the Concrete distribution. Intuitively, $p$ is trained to assign a high drop probability to the feature that is irrelevant to or redundant for predicting the target variable $Y$.

### 3.3 Deterministic Compressed Representation

With Drop-Bottleneck, one can simply obtain the deterministic version of the compressed representation as $\bar{Z} = \bar{C}_{\boldsymbol{p}}(X) = [\bar{c}(X_1, p_1), \ldots, \bar{c}(X_d, p_d)]$ for

$$\bar{c}(X_i, p_i) = \bar{b} \cdot \mathbb{1}(p_i < 0.5) \cdot X_i, \quad \text{where} \quad \bar{b} = \frac{d}{\sum_k \mathbb{1}(p_k < 0.5)}. \tag{10}$$

$\bar{b}$ is defined similarly to $b$ with a minor exception that the scaling is done based on the actual, deterministic number of the dropped features. The deterministic compressed representation $\bar{Z}$ is useful for inference tasks that require stability or consistency of the representation as well as reducing the feature dimensionality for inference, as we demonstrate in Section 5.4.

### 3.4 Training with Drop-Bottleneck

We present how Drop-Bottleneck (DB) is trained with the full IB objective allowing joint training with a feature extractor. Since DB proposes only a new compression term, any existing method for maximizing the prediction term $I(Z;Y)$ can be adopted. We below discuss an example with Deep Infomax (Hjelm et al., 2019) since our exploration method uses this framework (Section 4). Deep Infomax, instead of $I(Z;Y)$, maximizes its Jensen-Shannon mutual information estimator

$$I_{\psi}^{\text{JSD}}(Z;Y) = \frac{1}{2} \left( \mathbb{E}_{\mathbb{P}_{ZY}}[-\zeta(-T_{\psi}(Z,Y))] - \mathbb{E}_{\mathbb{P}_Z \otimes \mathbb{P}_Y}[\zeta(T_{\psi}(Z,\tilde{Y}))] + \log 4 \right), \tag{11}$$

where $T_{\psi}$ is a discriminator network with parameter $\psi$ and $\zeta(\cdot)$ is the softplus function.

Finally, the IB objective with Drop-Bottleneck becomes

$$\text{minimize} -I_{\psi}^{\text{JSD}}(Z;Y) + \beta \sum_{i=1}^{d} H(X_i)(1 - p_i), \tag{12}$$

which can be optimized via gradient descent. To make $\boldsymbol{p}$ more freely trainable, we suggest simple element-wise parameterization of $\boldsymbol{p}$ as $p_i = \sigma(p_i')$ and initializing $p_i' \sim \text{Uniform}(a, b)$. We obtain the input variable $X$ from a feature extractor, whose parameters are trained via the prediction term, jointly with $\boldsymbol{p}$ and $\psi$. In next section, we will discuss its application to hard exploration problems for reinforcement learning.

## 4 ROBUST EXPLORATION WITH DROP-BOTTLENECK

Based on DB, we propose an exploration method that is robust against highly noisy observations in a very sparse reward environment for reinforcement learning tasks. We first define a parametric feature extractor $f_\phi$ that maps a state to $X$. For transitions $(S, A, S')$ where $S, A$, and $S'$ are current states, actions and next states, respectively, we use the DB objective with

$$X = f_\phi(S'), \quad Z = C_{\boldsymbol{p}}(X), \quad Y = C_{\boldsymbol{p}}(f_\phi(S)). \tag{13}$$

For every transition $(S, A, S')$, the compression term $I(Z; X) = I(C_{\boldsymbol{p}}(f_\phi(S')); f_\phi(S'))$ encourages $C_{\boldsymbol{p}}$ to drop unnecessary features of the next state embedding $f_\phi(S')$ as possible. The prediction term $I(Z; Y) = I(C_{\boldsymbol{p}}(f_\phi(S')); C_{\boldsymbol{p}}(f_\phi(S)))$ makes the compressed representations of the current and next state embeddings, $C_{\boldsymbol{p}}(f_\phi(S))$ and $C_{\boldsymbol{p}}(f_\phi(S'))$, informative about each other.

Applying Equation (13) to Equation (12), the Drop-Bottleneck objective for exploration becomes

$$\text{minimize} -I_\psi^{\text{JSD}}(C_{\boldsymbol{p}}(f_\phi(S')); C_{\boldsymbol{p}}(f_\phi(S))) + \beta \sum_{i=1}^d H((f_\phi(S'))_i)(1 - p_i). \tag{14}$$

While $f_\phi$, $\boldsymbol{p}$, and $T_\psi$ are being trained online, we use them for the agent's exploration with the help of episodic memory inspired by Savinov et al. (2019). Starting from an empty episodic memory $M$, we add the learned feature of the observation at each step. For example, at time step $t$, the episodic memory is $M = \{\bar{C}_{\boldsymbol{p}}(f_\phi(s_1)), \bar{C}_{\boldsymbol{p}}(f_\phi(s_2)), \dots, \bar{C}_{\boldsymbol{p}}(f_\phi(s_{t-1}))\}$ where $s_1, \dots, s_{t-1}$ are the earlier observations from that episode. We then quantify the degree of novelty of a new observation as an intrinsic reward. Specifically, the intrinsic reward for $s_t$ is computed utilizing the Deep Infomax discriminator $T_\psi$, which is trained to output a large value for joint (or likely) input and a small value for marginal (or arbitrary) input:

$$r_{M,t}^{\text{i}}(s_t) = \frac{1}{t-1} \sum_{j=1}^{t-1} [g(s_t, s_j) + g(s_j, s_t)], \text{s.t. } g(x, y) = \zeta(-T_\psi(\bar{C}_{\boldsymbol{p}}(f_\phi(x)), \bar{C}_{\boldsymbol{p}}(f_\phi(y)))), \tag{15}$$

where $g(s_t, s_j)$ and $g(s_j, s_t)$ denote the unlikeliness of $s_t$ being the next and the previous state of $s_j$, respectively. Thus, intuitively, for $s_t$ that is close to a region covered by the earlier observations in the state space, $r_{M,t}^{\text{i}}(s_t)$ becomes low, and vice versa. It results in a solid exploration method capable of handling noisy environments with very sparse rewards. For computing the intrinsic reward, we use the DB's deterministic compressed representation of states to provide stable exploration signals to the policy optimization.

## 5 EXPERIMENTS

We carry out three types of experiments to evaluate Drop-Bottleneck (DB) in multiple aspects. First, we apply DB exploration to multiple VizDoom (Kempka et al., 2016) and DMLab (Beattie et al., 2016) environments with three hard noise settings, where we compare with state-of-the-art methods as well as its VIB variant. Second, we empirically show that DB is superior to VIB for adversarial robustness and feature dimensionality reduction in ImageNet classification (Russakovsky et al., 2015), and we juxtapose DB with VCEB, which employs a different form of the IB object, for the same adversarial robustness test, in Appendix A. Finally, in Appendix B, we make another comparison with VIB in terms of removal of task-irrelevant information and the validity of the deterministic compressed representation, where Appendix C provides the visualization of the task-irrelevant information removal on the same task.

### 5.1 EXPERIMENTAL SETUP FOR EXPLORATION TASKS

For the exploration tasks on VizDoom (Kempka et al., 2016) and DMLab (Beattie et al., 2016), we use the proximal policy optimization (PPO) algorithm (Schulman et al., 2017) as the base RL

method. We employ ICM from Pathak et al. (2017), and EC and ECO from Savinov et al. (2019) as baseline exploration methods. ICM learns a dynamics model of the environment and uses the prediction errors as exploration signals for the agent. EC and ECO are curiosity methods that use episodic memory to produce novelty bonuses according to the reachability of new observations, and show the state-of-the-art exploration performance on VizDoom and DMLab navigation tasks. In summary, we compare with four baseline methods: PPO, PPO + ICM, PPO + EC, and PPO + ECO. Additionally, we report the performance of our method combined with VIB instead of DB.

We conduct experiments with three versions of noisy-TV settings named "Image Action", "Noise" and "Noise Action", as proposed in Savinov et al. (2019). We present more details of noise and their observation examples in Appendix E.1. Following Savinov et al. (2019), we resize observations as $160 \times 120$ only for the episodic curiosity module while as $84 \times 84$ for the PPO policy and exploration methods. We use the official source code[1] of Savinov et al. (2019) to implement and configure the baselines (ICM, EC and ECO) and the three noise settings.

For the feature extractor $f_\phi$, we use the same CNN with the policy network of PPO from Mnih et al. (2015). The only modification is to use $d = 128$ *i.e.* 128-dimensional features instead of $512$ to make features lightweight enough to be stored in the episodic memory. The Deep Infomax discriminator $T_\psi$ consists of three FC hidden layers with $64, 32, 16$ units each, followed by a final FC layer for prediction. We initialize the drop probability $\boldsymbol{p}$ with $p_i = \sigma(p_i')$ and $p_i' \sim \text{Uniform}(a, b)$ where $a = -2, b = 1$. We collect samples and update $\boldsymbol{p}, T_\psi, f_\phi$ with Equation (14) every 10.8K and 21.6K time steps in VizDoom and DMLab, respectively, with a batch size of 512. We duplicate the compressed representation 50 times with differently sampled drop masks, which help better training of the feature extractor, the drop probability and the discriminator by forming diverse subsets of features. Please refer to Appendix E for more details of our experimental setup.

## 5.2    EXPLORATION IN NOISY STATIC MAZE ENVIRONMENTS

VizDoom (Kempka et al., 2016) provides a static 3D maze environment. We experiment on the *My-WayHome* task with nine different settings by combining three reward conditions ("Dense", "Sparse" and "Very Sparse") in Pathak et al. (2017) and three noise settings in the previous section. In the "Dense", "Sparse" and "Very Sparse" cases, the agent is randomly spawned in a near, medium and very distant room, respectively. Thus, "Dense" is a relatively easy task for the agent to reach the goal even with a short random walk, while "Sparse" and "Very Sparse" require the agent to perform a series of directed actions, which make the goal-oriented navigation difficult.

Table 1 and Figure 1 compare the DB exploration with three baselines, PPO, PPO + ICM, and PPO + ECO on the VizDoom tasks, in terms of the final performance and how quickly they learn. The results suggest that even in the static maze environments, the three noise settings can degrade the performance of the exploration by large margins. On the other hand, our method with DB shows robustness to such noise or task-irrelevant information, and outperforms the baselines in all the nine challenging tasks, whereas our method combined with VIB does not exhibit competitive results.

## 5.3    EXPLORATION IN NOISY AND RANDOMLY GENERATED MAZE ENVIRONMENTS

As a more challenging exploration task, we employ DMLab (Savinov et al., 2019), which are general and randomly generated maze environments where at the beginning of every episode, each maze is procedurally generated with its random goal location. Thanks to the random map generator, each method is evaluated on test mazes that are independent of training mazes. As done in Savinov et al. (2019), we test with six settings according to two reward conditions ("Sparse" and "Very Sparse") and the three noise settings. In the "Sparse" scenario, the agent is (re-)spawned at a random location when each episode begins or every time it reaches the goal *i.e.* the sparse reward; the agent should reach the goal as many times as possible within the fixed episode length. The "Very Sparse" is its harder version: the agent does not get (re-)spawned near or in the same room with the goal.

Table 1 compares between different exploration methods on the DMLab tasks. The results demonstrate that our DB exploration method achieves state-of-the-art performance with significant margins from the baselines on all the 6 tasks, and performs better than our method equipped with VIB as well. The plots too suggest that our method provides stable exploration signals to the agent under different environmental and noise settings. As mentioned in Section 5.1, our method can achieve better per-

---

[1]https://github.com/google-research/episodic-curiosity.

Table 1: Comparison of the average episodic sum of rewards in VizDoom (over 10 runs) and DMLab (over 30 runs) under three noise settings: Image Action (IA), Noise (N) and Noise Action (NA). The values are measured after 6M and 20M (4 action-repeated) steps for VizDoom and DMLab respectively, with no seed tuning done. Baseline results for DMLab are cited from Savinov et al. (2019). Grid Oracle[†] provides the performance upper bound by the oracle method for DMLab tasks.

| Method | VizDoom | | | | | | | | | DMLab | | | | | |
| | Dense | | | Sparse | | | Very Sparse | | | Sparse | | | Very Sparse | | |
| | IA | N | NA | IA | N | NA | IA | N | NA | IA | N | NA | IA | N | NA |
|---|---|---|---|---|---|---|---|---|---|---|---|---|---|---|---|
| PPO (Schulman et al., 2017) | **1.00** | **1.00** | **1.00** | 0.00 | 0.00 | 0.00 | 0.00 | 0.00 | 0.00 | 8.5 | 11.6 | 9.8 | 6.3 | 8.7 | 6.1 |
| PPO + ICM (Pathak et al., 2017) | 0.87 | **1.00** | **1.00** | 0.00 | 0.50 | 0.40 | 0.00 | 0.73 | 0.20 | 6.9 | 7.7 | 7.6 | 4.9 | 6.0 | 5.7 |
| PPO + EC (Savinov et al., 2019) | – | – | – | – | – | – | – | – | – | 13.1 | 18.7 | 14.8 | 7.4 | 13.4 | 11.3 |
| PPO + ECO (Savinov et al., 2019) | 0.71 | 0.81 | 0.72 | 0.21 | 0.70 | 0.33 | 0.19 | 0.79 | 0.50 | 18.5 | 28.2 | 18.9 | 16.8 | 26.0 | 12.5 |
| PPO + Ours (VIB) | **1.00** | **1.00** | **1.00** | 0.21 | 0.61 | 0.40 | 0.37 | 0.70 | 0.67 | 28.2 | 31.9 | 27.1 | 23.5 | 25.4 | 22.3 |
| PPO + Ours (Drop-Bottleneck) | **1.00** | **1.00** | **1.00** | **0.90** | **1.00** | **0.99** | **0.90** | **1.00** | **0.90** | **30.4** | **32.7** | **30.6** | **28.8** | **29.1** | **26.9** |
| PPO + Grid Oracle[†] | – | – | – | – | – | – | – | – | – | 37.4 | 38.8 | 39.3 | 36.3 | 35.5 | 35.4 |

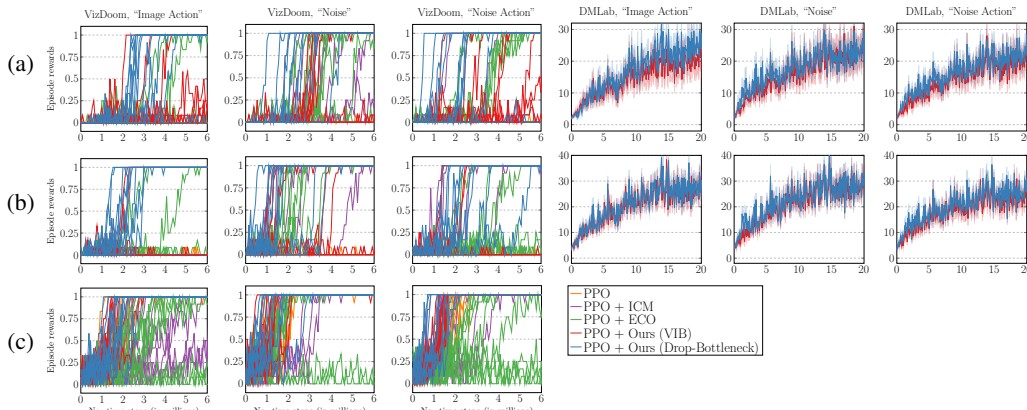

Figure 1: Reward trajectories as a function of training step (in millions) for VizDoom (columns 1-3) and DMLab (columns 4-6) with (a) Very Sparse, (b) Sparse and (c) Dense settings. For VizDoom tasks, we show all the 10 runs per method. For DMLab tasks, we show the averaged episode rewards over 30 runs of our exploration with the 95% confidence intervals.

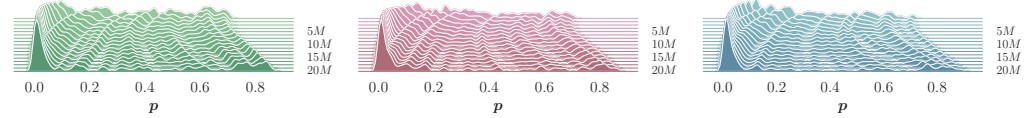

Figure 2: Evolution examples of the drop probability distribution $p$ on Very Sparse DMLab environments with (left) Image Action, (middle) Noise and (right) Noise Action settings. Each figure shows a histogram per $p$ value according to training iterations (the more front is the more recent).

formance even using much lower resolution observations of $84 \times 84$ than $160 \times 120$ of EC and ECO. Also, excluding the policy network, our method maintains 0.5M parameters, which is significantly smaller compared to ECO with 13M parameters. Please refer to Appendix D for an ablation study,

Figure 2 shows evolution examples of the drop probability distribution over training time steps. It overviews the role of drop probability $p$ in DB. As the joint training of the feature extractor $f_\phi$ with $p$ progresses, $p$ gets separated into high- and low-value groups, where the former drops task-irrelevant or redundant features and the latter preserves task-relevant features. This suggests that in the DMLab environments, the DB objective of Equation (14) successfully encourages dropping the features unrelated to transition between observations and also the deterministic compressed representation becomes stable as the training progresses.

Table 2: Results of the adversarial robustness for Drop-Bottleneck (DB) and Variational Information Bottleneck (VIB) (Alemi et al., 2017) with the targeted $\ell_2$ and $\ell_\infty$ attacks (Carlini & Wagner, 2017). *Succ.* denotes the attack success rate in % (lower is better), and *Dist.* is the average perturbation distance over successful adversarial examples (higher is better).

| Attack type | $\beta$ | VIB | | DB | | DB (determ.) | |
|---|---|---|---|---|---|---|---|
| | | Succ. | Dist. | Succ. | Dist. | Succ. | Dist. |
| $\ell_2$ | 0.0001 | 99.5 | 0.806 | 100.0 | 0.929 | 99.5 | 0.923 |
| | 0.0003162 | 99.5 | 0.751 | 100.0 | 0.944 | 100.0 | 0.941 |
| | 0.001 | 100.0 | 0.796 | 99.5 | 1.097 | 100.0 | 1.134 |
| | 0.003162 | 99.5 | 0.842 | 27.0 | 3.434 | 23.0 | 2.565 |
| | 0.01 | 100.0 | 0.936 | 18.5 | 6.847 | 20.0 | 6.551 |
| | 0.03162 | 100.0 | 0.695 | 41.0 | 2.160 | 39.5 | 1.953 |
| | 0.1 | 99.5 | 0.874 | 85.5 | 2.850 | 85.5 | 2.348 |
| $\ell_\infty$ | 0.0001 | 99.5 | 0.015 | 91.0 | 0.013 | 95.5 | 0.009 |
| | 0.0003162 | 99.5 | 0.017 | 85.0 | 0.016 | 91.5 | 0.009 |
| | 0.001 | 100.0 | 0.017 | 62.5 | 0.020 | 70.0 | 0.012 |
| | 0.003162 | 97.5 | 0.017 | 1.5 | 0.009 | 1.5 | 0.020 |
| | 0.01 | 87.0 | 0.019 | 2.0 | 0.022 | 2.0 | 0.013 |
| | 0.03162 | 25.0 | 0.121 | 8.5 | 0.022 | 8.0 | 0.023 |
| | 0.1 | 15.5 | 0.202 | 23.0 | 0.017 | 23.0 | 0.019 |

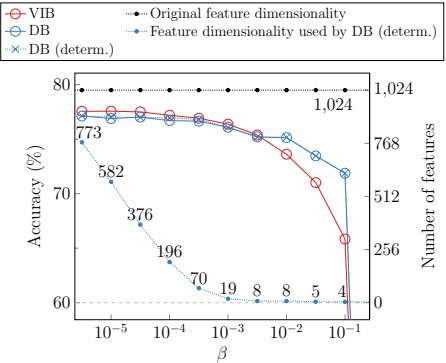

Figure 3: Classification accuracy of Inception-ResNet-v2 equipped with VIB (Alemi et al., 2017) and DB on ImageNet validation set (Russakovsky et al., 2015). DB (determ.) quickly drops many feature dimensions with increased $\beta$, while VIB retains them at 1024 regardless of $\beta$.

## 5.4 COMPARISON WITH VIB: ADVERSARIAL ROBUSTNESS & DIMENSION REDUCTION

We experiment with image classification on ImageNet (Russakovsky et al., 2015) to compare Drop-Bottleneck (DB) with Variational Information Bottleneck (VIB) (Alemi et al., 2017), the most widely-used IB framework, regarding the robustness to adversarial attacks and the reduction of feature dimensionality. We follow the experimental setup from Alemi et al. (2017) with some exceptions. We use $\beta_1 = 0.9$ and no learning rate decay for DB's Adam optimizer (Kingma & Ba, 2015). For prediction, we use one Monte Carlo sample of each stochastic representation. Additionally, we provide a similar comparison with the mutual information-based feature selection method.

**Robustness to adversarial attacks**. Following Alemi et al. (2017), we employ the targeted $\ell_2$ and $\ell_\infty$ adversarial attacks from Carlini & Wagner (2017). For each method, we determine the first 200 validation images on ImageNet that are classified correctly, and apply the attacks to them by selecting uniformly random attack target classes. Please refer to Appendix E.2 for further details.

Table 2 shows the results. For the targeted $\ell_2$ attacks, choosing the value of $\beta$ from $[0.003162, 0.1]$ provides the improved robustness of DB with the maximum at $\beta = 0.01$. On the other hand, VIB has no improved robustness in all ranges of $\beta$. For the targeted $\ell_\infty$ attacks, DB can reduce the attack success rate even near to $0\%$ (*e.g.* $\beta = 0.003162$ or $0.01$). Although VIB decreases the attack success rate to $15.5\%$ at $\beta = 0.1$, VIB already suffers from the performance degradation at $\beta = 0.1$ compared to DB (Figure 3), and increasing $\beta$ accelerates VIB's degradation even further. Note that the validation accuracies of both VIB and DB are close to zero at $\beta = 0.3162$.

**Dimensionality reduction**. Figure 3 compares the accuracy of DB and VIB by varying $\beta$ on the ImageNet validation set. Overall, their accuracies develop similarly with respect to $\beta$; while VIB is slightly better in the lower range of $\beta$, DB produces better accuracy in the higher range of $\beta$. Note that *DB (determ.)* shows the almost identical accuracy plot with *DB*. Importantly, *DB (determ.)* still achieves a reasonable validation accuracy ($\geq 75\%$) using only a few feature dimensions (*e.g.* 8) out of the original 1024 dimensions. This suggests that DB's deterministic compressed representation can greatly reduce the feature dimensionality for inference with only a small trade-off with the performance. It is useful for improving the efficiency of the model after the training is complete. On the other hand, VIB has no such capability. Finally, as Figure 3 shows, the trade-off between the dimensionality reduction and the performance can be controlled by the value of $\beta$.

**Comparison with feature selection**. As the deterministic representation of DB, *DB (determ.)*, provides the dimensionality reduction, we also empirically compare DB with the univariate mutual information-based feature selection method for obtaining the feature space with a reduced dimensionality. In the experiments, the same features provided to DB and VIB are given as input to the feature selection method as well, and for a more straightforward comparison, we let the feature se-

Table 3: Results of the adversarial robustness for Drop-Bottleneck (DB) and the mutual information-based feature selection with the targeted $\ell_2$ and $\ell_\infty$ attacks (Carlini & Wagner, 2017), using the same number of features. *Succ.* denotes the attack success rate in % (lower is better), and *Dist.* is the average perturbation distance over successful adversarial examples (higher is better).

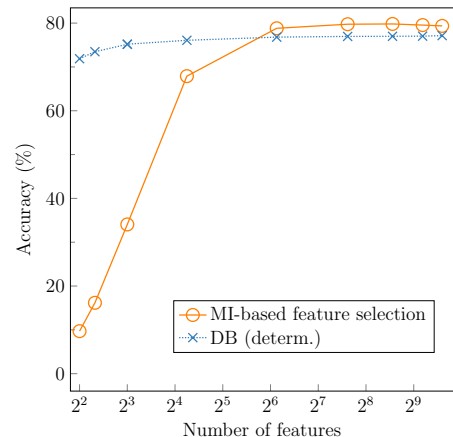

| Attack type | # of features | MI-based FS | | DB (determ.) | |
|---|---|---|---|---|---|
| | | Succ. | Dist. | Succ. | Dist. |
| $\ell_2$ | 196 | 99.5 | 1.484 | 99.5 | 0.923 |
| | 70 | 100.0 | 1.323 | 100.0 | 0.941 |
| | 19 | 99.5 | 1.161 | 100.0 | 1.134 |
| | 8 | 99.5 | 1.164 | 20.0 | 6.551 |
| | 5 | 97.0 | 1.202 | 39.5 | 1.953 |
| | 4 | 97.0 | 1.127 | 85.5 | 2.348 |
| $\ell_\infty$ | 196 | 99.5 | 0.016 | 95.5 | 0.009 |
| | 70 | 100.0 | 0.014 | 91.5 | 0.009 |
| | 19 | 99.5 | 0.013 | 70.0 | 0.012 |
| | 8 | 99.5 | 0.014 | 2.0 | 0.013 |
| | 5 | 97.0 | 0.016 | 8.0 | 0.023 |
| | 4 | 97.0 | 0.015 | 23.0 | 0.019 |

Figure 4: Classification accuracy of Inception-ResNet-v2 equipped with the mutual information-based feature selection and DB on ImageNet validation set (Russakovsky et al., 2015), using the same number of features.

lection method preserve the same number of features as *DB (determ.)*. Refer to Appendix E.3 for further details of the feature selection procedure. Figure 4 shows the classification accuracy of the two methods for the same numbers of features. The results show that while the mutual information-based feature selection method could provide a marginal performance benefit when a larger subset of the pre-trained features is preserved, DB is significantly better at retaining the accuracy with a small number of feature dimensions. For instance, DB achieves the accuracy over $71\%$ even with 4 features, but the accuracy of feature selection method drops from $\approx 68\%$ to $\approx 10\%$ when the number of features is $< 2^6$. Also, we make a comparison of the adversarial robustness; Table 3 suggests that the features preserved with the feature selection method show almost no robustness to the targeted $\ell_2$ and $\ell_\infty$ attacks, where every attack success rate is $\geq 97\%$. On the other hand, *DB (determ.)* can reduce the success rate to $20\%$ for the $\ell_2$ and to $2\%$ for the $\ell_\infty$ attacks with 8 features.

## 6 CONCLUSION

We presented Drop-Bottleneck as a novel information bottleneck method where compression is done by discretely dropping input features, taking into account each input feature's relevance to the target variable and allowing its joint training with a feature extractor. We then proposed an exploration method based on Drop-Bottleneck, and it showed state-of-the-art performance on multiple noisy and reward-sparse navigation environments from VizDoom and DMLab. The results showed the robustness of Drop-Bottleneck's compressed representation against noise or task-irrelevant information. With experiments on ImageNet, we also showed that Drop-Bottleneck achieves better adversarial robustness compared to VIB and can reduce the feature dimension for inference. In the exploration experiments, we directly fed the noisy observations to the policy, which can be one source of performance degradation in noisy environments. Therefore, applying Drop-Bottleneck to the policy network can improve its generalization further, which will be one interesting future research.

### ACKNOWLEDGMENTS

We thank Myeongjang Pyeon, Hyunwoo Kim, Byeongchang Kim and the anonymous reviewers for the helpful comments. This work was supported by Samsung Advanced Institute of Technology, Basic Science Research Program through the National Research Foundation of Korea (NRF) (2020R1A2B5B03095585) and Institute of Information & communications Technology Planning & Evaluation (IITP) grant funded by the Korea government (MSIT) (No.2019-0-01082, SW StarLab). Jaekyeom Kim was supported by Hyundai Motor Chung Mong-Koo Foundation. Gunhee Kim is the corresponding author.

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

## A    COMPARISON WITH VCEB: ADVERSARIAL ROBUSTNESS

In this section, we compare Drop-Bottleneck (DB) with Variational Conditional Entropy Bottleneck (VCEB) (Fischer, 2020; Fischer & Alemi, 2020) on the same ImageNet (Russakovsky et al., 2015) tasks for the adversarial robustness as in Section 5.4. VCEB variationally approximates the CEB objective, which is defined as

$$\text{minimize} -I(Z;Y) + \gamma I(Z;X|Y). \tag{16}$$

Note that Equation (16) is an alternative form of the original IB objective, Equation (1), as $I(Z;X,Y) = I(Z;Y) + I(Z;X|Y) = I(Z;X) + I(Z;Y|X)$ and $I(Z;Y|X) = 0$ $(\because Z \perp\!\!\!\perp Y|X)$. As in Section 5.4, we employ the experimental setup from VIB (Alemi et al., 2017) with small modifications to the hyperparameters for the Adam optimizer (Kingma & Ba, 2015) that $\beta_1 = 0.9$ and no learning rate decay is used. Additionally for VCEB, we apply the configurations suggested by Fischer & Alemi (2020): 1) at test time, use the mean of the Gaussian encoding instead of sampling from the distribution, and 2) reparameterize $\gamma = \exp(-\rho)$ and anneal the value of $\rho$ from $\rho = 100$ to the final $\rho$ during training. For our experiments, the annealing is performed via the first $100000$ training steps, where each epoch consists of $6405$ steps.

Employing the experimental setup from Alemi et al. (2017), we adopt the targeted $\ell_2$ and $\ell_\infty$ adversarial attacks from Carlini & Wagner (2017). We determine the first 200 correctly classified validation images on ImageNet for each setting and perform the attacks where the attack target classes are chosen randomly and uniformly. Further details are described in Appendix E.2.

Figure 5 visualizes the classification accuracy for each corresponding final value of $\rho$, and the adversarial robustness results are shown in Table 4. Overall, both VCEB and DB provide meaningful robustness to the targeted $\ell_2$ and $\ell_\infty$ attacks. For the targeted $\ell_2$ attacks, although VCEB could achieve the higher average perturbation distance over successful attacks, DB and its deterministic form show better robustness compared to VCEB in terms of the attack success rates: $18.5\%$ (*DB* at $\beta = 0.01$) and $20.0\%$ (*DB (determ.)* at $\beta = 0.01$) versus $45.0\%$ (VCEB at the final $\rho = 3.454$). For the targeted $\ell_\infty$ attacks, DB and its deterministic version again achieve the lower attack success rates than VCEB: $1.5\%$ and $2.0\%$ (*DB* and *DB (determ.)* at $\beta \in \{0.003162, 0.01\}$)) versus $12.5\%$ (VCEB at the final $\rho = 3.454$).

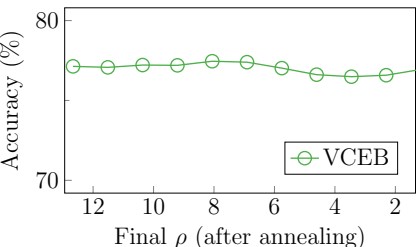

Figure 5: Classification accuracy of Inception-ResNet-v2 equipped with VCEB (Fischer, 2020; Fischer & Alemi, 2020) on ImageNet (Russakovsky et al., 2015). $\rho$ is annealed from 100 to the final $\rho$ over the first 100000 training steps.

## B    REMOVAL OF TASK-IRRELEVANT INFORMATION AND VALIDITY OF DETERMINISTIC COMPRESSED REPRESENTATION

We experiment Drop-Bottleneck (DB) and Variational Information Bottleneck (VIB) (Alemi et al., 2017) on occluded image classification tasks to show the following:

- DB can control the degree of compression (*i.e.* degree of removal of task-irrelevant information) in the same way with VIB as the popular IB method.
- DB's deterministic compressed representation works as a reasonable replacement for its stochastic compressed representation and it maintains the learned indistinguishability better than the attempt of VIB's deterministic compressed representation.

We employ the Occluded CIFAR dataset using the experimental settings from Achille & Soatto (2018). The Occluded CIFAR dataset is created by occluding CIFAR-10 (Krizhevsky, 2009) images with MNIST (LeCun et al., 2010) images as shown in Figure 6a, and each image has two labels of CIFAR and MNIST. We use a modified version of All-CNN-32 (Achille & Soatto, 2018) equipped with an IB method (either of DB or VIB) for the feature extractor whose output dimension is $d$. Each

Table 4: Results of the adversarial robustness for Drop-Bottleneck (DB) and Variational Conditional Entropy Bottleneck (VCEB) (Fischer, 2020; Fischer & Alemi, 2020) with the targeted $\ell_2$ and $\ell_\infty$ attacks (Carlini & Wagner, 2017). *Succ.* denotes the attack success rate in % (lower is better), and *Dist.* is the average perturbation distance over successful adversarial examples (higher is better). $^\dagger\rho$ for VCEB is annealed from 100 to the final $\rho$ over the first 100000 training steps.

| Attack type | Constraint on $I(Z; X\|Y)$ | | | Constraint on $I(Z; X)$ | | | | |
|---|---|---|---|---|---|---|---|---|
| | Final $\rho^\dagger$ | VCEB | | $\beta$ | DB | | DB (determ.) | |
| | | Succ. | Dist. | | Succ. | Dist. | Succ. | Dist. |
| $\ell_2$ | 9.210 | 99.0 | 1.200 | 0.0001 | 100.0 | 0.929 | 99.5 | 0.923 |
| | 8.059 | 92.0 | 2.028 | 0.0003162 | 100.0 | 0.944 | 100.0 | 0.941 |
| | 6.908 | 86.5 | 5.040 | 0.001 | 99.5 | 1.097 | 100.0 | 1.134 |
| | 5.757 | 65.0 | 7.198 | 0.003162 | 27.0 | 3.434 | 23.0 | 2.565 |
| | 4.605 | 46.0 | 12.016 | 0.01 | 18.5 | 6.847 | 20.0 | 6.551 |
| | 3.454 | 45.0 | 10.744 | 0.03162 | 41.0 | 2.160 | 39.5 | 1.953 |
| | 2.303 | 53.0 | 14.021 | 0.1 | 85.5 | 2.850 | 85.5 | 2.348 |
| $\ell_\infty$ | 9.210 | 86.0 | 0.012 | 0.0001 | 91.0 | 0.013 | 95.5 | 0.009 |
| | 8.059 | 64.5 | 0.013 | 0.0003162 | 85.0 | 0.016 | 91.5 | 0.009 |
| | 6.908 | 48.0 | 0.016 | 0.001 | 62.5 | 0.020 | 70.0 | 0.012 |
| | 5.757 | 29.0 | 0.019 | 0.003162 | 1.5 | 0.009 | 1.5 | 0.020 |
| | 4.605 | 17.0 | 0.025 | 0.01 | 2.0 | 0.022 | 2.0 | 0.013 |
| | 3.454 | 12.5 | 0.025 | 0.03162 | 8.5 | 0.022 | 8.0 | 0.023 |
| | 2.303 | 17.5 | 0.027 | 0.1 | 23.0 | 0.017 | 23.0 | 0.019 |

run consists of two phases. In the first phase, we train the feature extractor with a logistic classifier using both the classification loss for CIFAR and the compression objective of the IB method. Fixing the trained feature extractor, we train a logistic classifier for MNIST in the second phase. We train two different versions of classifiers for each of VIB and DB using stochastic or deterministic compressed representation from the feature extractor. For the deterministic representation of VIB, we use the mode of the output Gaussian distribution.

Figures 6b–6d contain the experimental results with $d = 32$, $d = 64$, and $d = 128$. In the first phase, DB retains only a subset of features that concentrate more on the CIFAR part of the images. Thus, the trained feature extractor preserves less information about the MNIST parts, and the errors of the MNIST classification are high. The first observation is that for both DB and VIB with the original stochastic compressed representation, *nuisance* plots show that increasing $\beta$ from the minimum value to $\sim 0.1/d$ barely changes the *primary* CIFAR errors but grows the *nuisance* MNIST errors up to $\sim 90\%$ (*i.e.* the maximum error for the 10-way classification). With even larger $\beta$, enforcing stronger compression results in the increase of the *primary* errors too, as shown in *primary* plots. This suggests that both DB and VIB provide fine controllability over the removal of task-irrelevant information.

Secondly, if we move our focus to the *nuisance (deterministic)* plots in Figures 6b–6d, which show the test errors on the *nuisance* MNIST classification with the feature extractor's deterministic representation, the results become different between DB and VIB. In DB, the *nuisance (deterministic)* plots follow the *nuisance* plots in the range of $\beta$ where the compression takes effect (*i.e.* where the *nuisance* errors increase). Moreover, the two plots get closer as $\beta$ increases. It means that Drop-Bottleneck's deterministic compressed representations maintain the majority of the indistinguishability for the task-irrelevant information learned during the first phase, especially when $\beta$ is large enough to enforce some degree of the compression. On the other hand, VIB's *nuisance (deterministic)* plots are largely different from the *nuisance* plots; even the *primary* errors rise before the *nuisance (deterministic)* errors reach their maximum values. This shows that employing the mode of VIB's output distribution as its deterministic representation results in loss of the learned indistinguishability.

In summary, we confirm that DB provides controllability over the degree of compression in a similar way as VIB. On the other hand, DB's deterministic representation can be a reasonable replacement

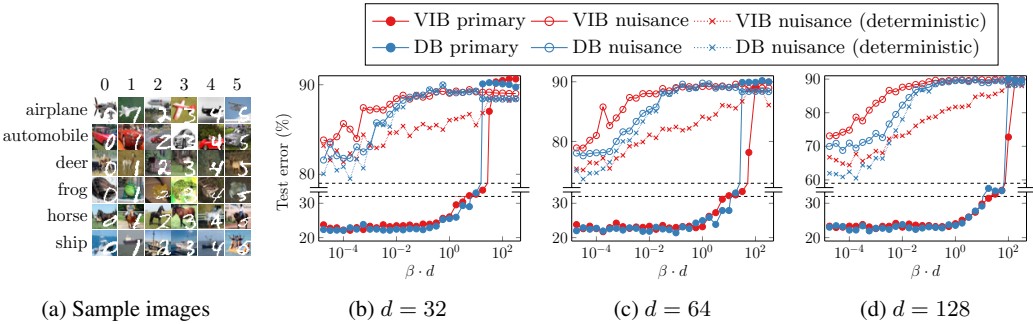

(a) Sample images        (b) $d = 32$        (c) $d = 64$        (d) $d = 128$

Figure 6: (a) A few samples from Occluded CIFAR dataset (Achille & Soatto, 2018). (b)–(d) Test error plots on the *primary* task (*i.e.* the classification of occluded CIFAR images) and on the *nuisance* tasks (*i.e.* classification of the MNIST digits). For all the three types of tasks, we use the same feature extractor trained for the *primary* task, where its deterministic representation is used only for training and test of the *nuisance (deterministic)* task.

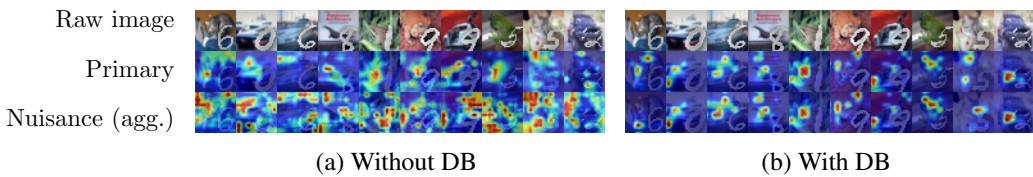

(a) Without DB             (b) With DB

Figure 7: Grad-CAM (Selvaraju et al., 2017) visualization for the last convolutional layer of the feature extractor on the Occluded CIFAR classification task. For the visualization, $d = 128$, and $\beta = 5.623/d$ for (b) are used. *Primary* denotes the maps of the logits for the primary labels. *Nuisance (agg.)* means the maps on the nuisance task aggregated over all the logits (*i.e.* 10 logits). (a) indicates that the feature extractor without DB trained on the primary task still outputs much information about the nuisance tasks, and thus the nuisance classifier could depend on the features extracted from the nuisance (MNIST) regions. In contrast, (b) suggests that the feature extractor with DB could learn to discard the nuisance features, so that the nuisance classifier mostly fails to learn due to the lack of nuisance-relevant features.

for its original stochastic representation in terms of preserving the learned indistinguishability, which is not exhibited by VIB.

## C    VISUALIZATION OF TASK-IRRELEVANT INFORMATION REMOVAL

In this section, we visualize the removal of task-irrelevant information with Drop-Bottleneck (DB). To this end, we employ the Occluded CIFAR dataset (Achille & Soatto, 2018) with the same experimental setup as in Appendix B. Each image of Occluded CIFAR dataset is one of the CIFAR-10 (Krizhevsky, 2009) images occluded by MNIST (LeCun et al., 2010) digit images. In the first phase of the experiments, the feature extractor and the classifier are trained on the primary (occluded CIFAR classification) task in a normal way. During the second phase, the learned feature extractor is fixed, and only a new classifier is trained on the nuisance (MNIST classification) task. In Appendix B, we quantitatively showed that the feature extractor with DB could focus more on the occluded CIFAR images and preserve less information about the MNIST occlusions. We take a qualitative approach in this section and visualize the phenomenon using Grad-CAM (Selvaraju et al., 2017). Grad-CAM is popular for providing visual explanation given convolutional neural networks with their target values (*e.g.* target logits in classification tasks).

We first sample multiple test images from the Occluded CIFAR dataset, and load full, trained models, which include the feature extractor, primary classifier and nuisance classifier. We then obtain the activation maps for the last convolutional layer of the feature extractor on the primary and nuisance tasks. On the primary task, we compute the activation maps simply targeting the logits for the sample labels. However, on the nuisance task, we get the activation maps of all the logits and aggregate them by taking the maximum of the maps at each pixel location. Therefore, the aggregated maps on the

nuisance task visualize the activation related to not only the logits for the true class labels but also the other logits, capturing most of the feature usage induced during the training on the nuisance task.

Figure 7 compares two trained models: the $d = 128$ model without DB, and the $d = 128$ model with DB (deterministic). We use the DB model with $\beta = 5.623/d$ for the visualization, as the value is sufficiently large enough to enforce strong compression while it still keeps the primary error not high. Figure 7a shows that regarding the logits for the nuisance (MNIST) task, a large portion of each image including the MNIST digit is activated in most cases, and thus it indicates that the feature extractor trained without DB preserves much of the nuisance features. On the other hand, Figure 7b visualizes that the feature extractor with DB outputs notably less of the nuisance features, preventing the nuisance classifier from learning correctly.

To sum up, we provide the visual demonstration that on the classification task with the Occluded CIFAR dataset, the feature extractor equipped with DB trained on the primary task could discard majority of the nuisance *i.e.* task-irrelevant information given a value of $\beta$ that is strong enough.

## D   ABLATION STUDY: EXPLORATION WITHOUT DROP-BOTTLENECK

We perform an ablation study to show Drop-Bottleneck (DB)'s ability of dealing with task-irrelevant input information. We examine the performance of the same exploration method as described in Section 4 but without DB; that is, the feature vectors are fully used with no dropping. In order to emphasize the effectiveness of DB for noisy and task-irrelevant information, we conduct experiments with both noisy and original (*i.e.* without explicit noise injection) settings.

Table 5: Comparison of the average episodic sum of rewards in DMLab tasks (over 30 runs), where PPO + Ours (No-Drop-Bottleneck) denotes our exploration method without DB. The original (*i.e.* without explicit noise injection) and three noisy settings are tested: Image Action (IA), Noise (N), Noise Action (NA) and Original (O). The values are measured after 20M (4 action-repeated) time steps, with no seed tuning done. Baseline results for DMLab are cited from Savinov et al. (2019).

| Method | DMLab | | | |
| --- | --- | --- | --- | --- |
| | Very Sparse | | | |
| | IA | N | NA | O |
| PPO (Schulman et al., 2017) | 6.3 | 8.7 | 6.1 | 8.6 |
| PPO + ICM (Pathak et al., 2017) | 4.9 | 6.0 | 5.7 | 11.2 |
| PPO + EC (Savinov et al., 2019) | 7.4 | 13.4 | 11.3 | 24.7 |
| PPO + ECO (Savinov et al., 2019) | 16.8 | 26.0 | 12.5 | 40.5 |
| PPO + Ours (No-Drop-Bottleneck) | 14.9 | 11.7 | 10.3 | 33.0 |
| PPO + Ours (Drop-Bottleneck) | **28.8** | **29.1** | **26.9** | **42.7** |
| Improvement (%) | 93.3 | 148.7 | 161.2 | 29.4 |

Table 5 shows the results on "Very Sparse" DMLab environments with "Image Action", "Noise", "Noise Action" and "Original" settings. Compared to "Original" setting where observations contain only implicit, inherent noisy information irrelevant to state transitions, DB brings much more significant improvement to exploration methods in "Image Action", "Noise", and "Noise Action" settings, which inject explicit, severe transition-irrelevant information to observations. These results suggest that DB plays an important role handling noisy or task-irrelevant input information.

## E   DETAILS OF EXPERIMENTS

We describe additional details of the experiments in Section 5. For all the experiments with DB, each entropy $H(X_i)$ is computed with the binning-based estimation using 32 bins, and the drop probability $\boldsymbol{p}$ is initialized with $p_i = \sigma(p_i')$ and $p_i' \sim \text{Uniform}(a, b)$ for $a = -2, b = 1$.

Table 6: Hyperparameters of PPO (Schulman et al., 2017), PPO + ICM (Pathak et al., 2017), PPO + ECO (Savinov et al., 2019), and PPO + Ours for the VizDoom experiments.

|  | PPO | PPO + ICM | PPO + ECO | PPO + Ours |
|---|---|---|---|---|
| PPO | | | | |
|     Learning rate | 0.00025 | 0.00025 | 0.00025 | 0.00025 |
|     Entropy coefficient | 0.01 | 0.01 | 0.01 | 0.01 |
|     Task reward scale | 5 | 5 | 5 | 5 |
| Exploration method | | | | |
|     Training period and sample size | – | 3K | 120K | 10.8K |
|     # of optimization epochs | – | 4 | 10 | 4 |
|     Intrinsic bonus scale | – | 0.01 | 1 | 0.001 |

Table 7: Hyperparameters of PPO (Schulman et al., 2017), PPO + ICM (Pathak et al., 2017), PPO + EC/ECO (Savinov et al., 2019), and PPO + Ours for the DMLab experiments.

|  | PPO | PPO + ICM | PPO + EC/ECO | PPO + Ours |
|---|---|---|---|---|
| PPO | | | | |
|     Learning rate | 0.00019 | 0.00025 | 0.00025 | 0.00025 |
|     Entropy coefficient | 0.0011 | 0.0042 | 0.0021 | 0.0011 |
|     Task reward scale | 1 | 1 | 1 | 1 |
| Exploration method | | | | |
|     Training period and sample size | – | 3K | 720K (ECO) | 21.6K |
|     # of optimization epochs | – | 4 | 10 (ECO) | 2 |
|     Intrinsic bonus scale | – | 0.55 | 0.030 | 0.005 |

### E.1 DETAILS OF EXPLORATION EXPERIMENTS

To supplement Section 5.1, we describe additional details of the VizDoom (Kempka et al., 2016) and DMLab (Beattie et al., 2016) exploration experiments. We collect training samples in a buffer and update $p, T_\psi, f_\phi$ with Equation (14) periodically. We use Adam optimizer (Kingma & Ba, 2015) with a learning rate of 0.0001 and a batch size of 512. In each optimization epoch, the training samples from the buffer are re-shuffled. For each mini-batch, we optimize the Deep Infomax (Hjelm et al., 2019) discriminator $T_\psi$ with 8 extra epochs with the same samples, to make the Jensen-Shannon mutual information bound tighter. This way of training $T_\psi$ only runs forward and backward passes on $T_\psi$ for the fixed output of the feature extractor $f_\phi$, and thus can be done with low computational cost. $\beta$ is the hyperparameter that determines the relative scales of the compression term and the Deep Infomax Jensen-Shannon mutual information estimator. It is tuned to $\beta = 0.001/128$ for DB and $\beta = 0.0005/128$ for VIB. To make experiments simpler, we normalize our intrinsic rewards with the running mean and standard deviation.

Table 6 and Table 7 report the hyperparameters of the methods for VizDoom and DMLab experiments, respectively. We tune the hyperparameters based on the ones provided by Savinov et al. (2019). Unless specified, we use the same hyperparameters with Savinov et al. (2019).

Under the three noise settings suggested in Savinov et al. (2019), the lower right quadrant of every observation is occupied by a TV screen as follows.

- "Image Action": Every time the agent performs a specific action, it changes the channel of the TV randomly to one of the 30 predefined animal images.

- "Noise": At every observation, a new noise pattern is sampled and shown on the TV screen (independently from the agent's actions).

- "Noise Action": Same as "Noise", but the noise pattern only changes when the agent does a specific action.

Figure 8 shows some observation examples from VizDoom and DMLab environments with "Image Action" and "Noise" settings.

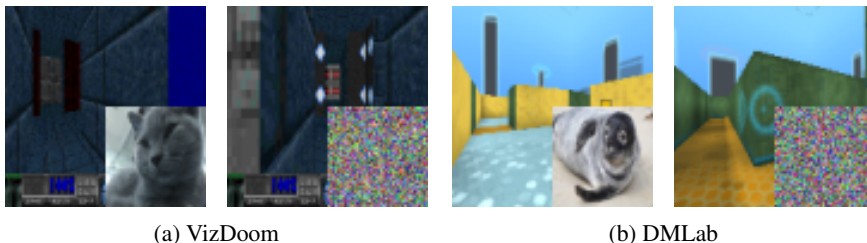

(a) VizDoom                    (b) DMLab

Figure 8: Example observations from VizDoom (Kempka et al., 2016) and DMLab (Beattie et al., 2016) environments with "Image Action" (first) and "Noise" (second) settings.

### E.2    DETAILS OF ADVERSARIAL ROBUSTNESS EXPERIMENTS

For the experiments, each input image is sized as $299 \times 299 \times 3$ and each pixel value is ranged $[-1, 1]$. We perform our adversarial robustness experiments based on the official source code[2] of Carlini & Wagner (2017). For the targeted $\ell_2$ attack we increase the number of binary search steps to 20 and use a batch size of 25. Other than the two, the default hyperparameters are used.

### E.3    DETAILS OF RUNNING FEATURE SELECTION METHOD

The input of the feature selection method is the same as DB's and VIB's: the input features are obtained with a pre-trained model of Inception-ResNet-v2 on ImageNet (Russakovsky et al., 2015). We randomly pick $100k$ samples out of the total $1281167$ training samples of ImageNet, and compute the relevance scores for features using those samples by estimating the mutual information between each feature and its label using the entropy estimation with the $k$-nearest neighbors (Kraskov et al., 2004; Ross, 2014). Given the computed scores, we perform the feature selection by preserving the features with the highest scores.

### E.4    DETAILS OF OCCLUDED IMAGE CLASSIFICATION EXPERIMENTS

We use a modified version of All-CNN-32 (Achille & Soatto, 2018). The model architecture for feature dimension $d$ is described in Table 8. Batch normalization (Ioffe & Szegedy, 2015) is applied to Conv layers, and the ReLU (Nair & Hinton, 2010; Glorot et al., 2011) activation is used at every hidden layer. We use the Adam optimizer (Kingma & Ba, 2015) with a learning rate of 0.001. To ensure the convergence, in each training, the model is trained for 200 epochs with a batch size of 100.

Table 8: The network architecture for the occluded image classification experiments.

| | | |
|---|---|---|
| | Input image $32 \times 32 \times 3$ | |
| Feature extractor | Conv [$3 \times 3$, 96, stride 1]
Conv [$3 \times 3$, 96, stride 1]
Conv [$3 \times 3$, 96, stride 2] | |
| | Conv [$3 \times 3$, 192, stride 1]
Conv [$3 \times 3$, 192, stride 1]
Conv [$3 \times 3$, 192, stride 2] | |
| | FC [$d$] | FC [$2d$] |
| | DB [$d$] | VIB [$d$] |
| Classifier | FC [10] | |
| | softmax | |

---

[2]https://github.com/carlini/nn_robust_attacks.

