# OpenReview forum: "Drop-Bottleneck: Learning Discrete Compressed Representation for Noise-Robust Exploration"
_ICLR.cc/2021/Conference — ICLR 2021 Poster_

### Official Review · AnonReviewer1 · 2020-10-21
**I am leaning towards recommending to reject this paper, mainly because the focus on the noisy TV problem is very narrow and the additional experiments on ImageNet do not fit to the paper's focus. Furthermore, the work does not discuss its relation to other feature selection works. However, the ImageNet experiments show that the presented idea can be useful.**

**Rating:** 6
**Confidence:** 4

**Review:**

Summary:
The paper contributes a novel method, Drop-Bottleneck (DB), for discretely dropping input features that are irrelevant for predicting the target variable. Key idea is to instantiate the compression term of the information bottleneck framework with learned term that sets irrelevant feature dimensions to 0. To this end, a drop probability is learned for each dimension. Dimensions that have a lower probability than 0.5 (a fixed threshold) of being relevant are set to 0.

Strong Points:
- The paper is well-written and easy to understand.
- Experiments show that DB works better than VIB in VizDoom and DMLab when a noisy-TV noise is added to the input images. Different noisy-TV noises are considered: changing the image when the agent performs an action, adding random noise to the TV, and adding random noise when the agent performs an action.
- Experiments also show that the obtained representation is more robust against l_2 and l_inf attacks in ImageNet. Furthermore, the experiments show that the approach can drop many ImageNet features while almost preserving the accuracy of a ResNet.
- The paper comes with code in the supplementary material.

Weak Points:
- I found the reinforcement learning experiments not convincing since only a fixed region of the input is modified by noise (ie. the noisy TV). Hence, the approach essentially identifies irrelevant pixel locations. Such a problem could be solved by a simple pre-processing step. The method won't work if the location of the noise changes. In general, limitations of the work are not discussed.
- The experiments on ImageNet are more interesting. However, the fact that individual dimensions (i.e. specific pixel locations) are identified as irrelevant is still a limiting factor. Furthermore, the experiments do not fit to the focus of this paper on reinforcement learning.
- The paper does not discuss connections of the presented approach to prior works for (discrete) feature selection. It only discusses connections to prior bottleneck methods.
- The paper does not perform experiments on datasets with meaningful features where a feature selection makes more sense than for specific pixels in images.

Additional feedback:
- It could be stated explicitly that H refers to the entropy. Currently, it is only implicitly defined in Eq. 6.
- I think it would be better to also cite Jang et. al (Categorical Reparameterization with Gumbel-Softmax) for the Concrete relaxation of the Bernoulli distribution.
- Figure 1 should either be improved or removed. I don't see much additional insights that can be gained from this figure.
- Of course, it would be very interesting to see if the drop probabilities correlate with the location of the noise inputs. It would be great if such an analysis could be added. This could replace Figure 1.

---

> ### Author Response · Authors · 2020-11-19
> **Author Response to AnonReviewer1**
>
> We appreciate your valuable comments. We will clarify all the concerns in our final draft and make our code public.
>
> 1. **RL experiments are not convincing - only a fixed region of the input is modified by noise (the noisy TV task).**
>
>   A. We adopt the experimental setup from the Savinov et al. (2019) since it is one of the most popular benchmarks for RL exploration in noisy and sparse-reward environments. Although the reviewer's comment is correct that the TV screen location is fixed, this setup has been widely used because RL agents have no supervision to simply ignore the fixed region and are easily fooled by the noisy TV even at fixed location, especially when its content is changed based on the agents' actions (Burda et al., 2019a;b; Kim et al., 2019; Savinov et al., 2019). Nonetheless, as suggested, if we let the location of the noisy TV not be fixed, the task will surely be harder. Please excuse that it is extremely hard to obtain the results in this setting during the rebuttal period, because no previous work has used this setting and reproducing all SOTA models in new environments is highly challenging and time-consuming.
>
>
> 2. **Individual dimensions (i.e. specific pixel locations) are identified as irrelevant & Experiments on datasets with meaningful features**
>
>   A. We clarify that our experiments are not equivalent to simply identifying noisy or irrelevant pixel locations in the images. The input to DB $X$ is not the raw input image but the output of a feature extractor (e.g. $f_\phi$ in Eq.(13)), which is jointly trained with the learnable drop probability (Section 3.4).
> In the ImageNet experiments, for example, the feature extractor takes the features from pre-trained Inception-ResNet-V2 as input, following the setup suggested by VIB (Alemi et al., 2017) (Section 5.4). In summary, DB can perform both feature learning and feature selection jointly.
>
>
> 3. **ImageNet experiments do not fit to the focus of this paper on reinforcement learning**
>
>   A. Since DB is a new information bottleneck (IB) framework, it is applicable to other tasks that IB can handle. The RL exploration tasks are our main focus but adversarial robustness in image classification is another interesting task to show the effectiveness of DB.
>
>
> 4. **Connections of the presented approach to prior works for (discrete) feature selection**
>
>   A. DB shares a similar idea with feature selection methods to preserve an important subset of feature variables. However, one important difference is that DB allows the joint training of the drop probability and the feature extractor that outputs $X$. Moreover, DB can provide improved results compared to the classic feature selection methods in terms of dimensionality reduction and providing adversarial robustness.
> We have tested and reported the results of other methods (mutual information-based feature selection and L1 regularization based method). For its details, please refer to our response to AnonReviewer3’s Q1 (L1), AnonReviewer4’s Q1 (MI), and the updated results in Section 5.4 of our new revision.
>
>
> 5. **Additional feedback**
>
>   A. We appreciate the additional feedback as well and we happily respond as follows:
> - We updated the draft to mention that $H$ denotes the entropy at the first use of the notation (right after Eq.(5)).
> - We added a citation to Jang et al. (Categorical Reparameterization with Gumbel-Softmax) for the Concrete relaxation of the Bernoulli distribution.
> - With Figure 1, we tried to visualize the evolution of episode rewards, since the values reported in Table 1 are the final values and do not show how quickly each method learns, especially for the VizDoom tasks. We updated Section 5.2 to state what can be checked with Figure 1.
> - As $X$ is obtained with the feature extractor, the drop probability and individual pixels are not directly correlated, which makes such analysis difficult. We will try to find other ways for better analysis and to include the results in the camera-ready version.

---

### Official Review · AnonReviewer3 · 2020-10-29
**A simple yet efficient idea for a new IB objective that works well across the board**

**Rating:** 7
**Confidence:** 4

**Review:**

The paper proposes a new Information Bottleneck objective, which compresses the latent by learning to drop features similar to DropOut. Unlike DropOut, a different probability is learnt for each latent feature/dimension using Concrete Relaxation. The experiments show that the works well.

Overall, I score this paper as an accept. While the approach is limited to dropping input features, which does not make it a general IB objective, it seems to work very well in the presented RL experiments as well as show robustness that is better than DVIB’s. Moreover, the paper is clearly written and engaging.

### Strengths

The paper proposes a simple yet effective idea. Using a Concrete Relaxation to learn DropOut probabilities has been done before, but the idea to have a separate probability per latent dimension is novel. The Drop-Bottleneck objective works directly on the input/latent layer, which means that the compression objective is easy to compute. This is nice because IB terms are usually cumbersome to compute. (However, this also requires the input/latent to already be disentangled, as dropping out features is limited in its expressiveness.)

The RL experiments on VizDoom and DMLab are convincing as are the ones on ImageNet. The additional experiment on the Occluded CIFAR-10 dataset in the appendix is also well thought-out and shows the advantage of this straightforward method over DVIB.

Moreover, once trained, features can be dropped out deterministically if so required, which allows for proper compression and consistent behaviour.

### Questions

Given the simple conceptual idea, this reviewer would be interested to see an ablation with using other methods of enforcing sparsity in the latent. Could L1 regularization of the latent activations be used instead of the $I[Z; X]$ term?

DB cannot provide the same generality as other IB objectives: the input (latent) has to be sufficiently disentangled already as the objective itself does not encourage further disentanglement by itself. Do the ImageNet experiments use extracted/pre-trained embeddings as latents?

---
### Rebuttal

I thank the authors for their reply. I'm more confident this is a good paper now.

---

> ### Author Response · Authors · 2020-11-19
> **Author Response to AnonReviewer3**
>
> We appreciate your valuable comments. We will clarify all the concerns in our final draft and make our code public.
>
> 1. **Other sparsity-enforcing methods such as L1 regularization of the latent activations**
>
>   A. As the review pointed out, the L1 regularization of the latent activations can make each latent vector sparser. On the other hand, DB's compression term also encourages dropping of feature variables and thus ultimately introduces sparsity to the latent space. One notable benefit of DB’s sparsity of the “latent space” over L1’s sparsity of “each latent vector” is that it can perform the feature dimensionality reduction too for improving the efficiency of the model.
> Moreover, empirically the L1 regularization is not as good as DB. We tested the L1 regularization to the latent activations (i.e. $X$ for DB) on ImageNet classification from Section 5.4. When the regularization coefficient is $\leq 0.000025$, its test accuracy is around $[74\\%,76\\%]$ (smaller than DB's accuracy with $\beta \leq 0.001$), and the accuracy quickly drops near to zero at larger coefficient values.
> We have tested and reported the results of another feature selection method (mutual information-based one). For its details, please refer to our response to AnonReviewer4’s Q1 and the updated results in Section 5.4.
>
>
> 2. **DB’s generality as other IB objectives - What if the input should already be disentangled?**
>
>   A. We jointly train the learnable drop probability and the feature extractor that outputs $X$ (e.g. $f_\phi$ in Eq.(13)). It makes DB less dependent on raw input data because it can learn optimal embeddings for DB together. Also, since DB encourages dropping of redundant features from $X$, as the training progresses, $Z$ becomes likely to preserve less correlated features.
>
>
> 3. **Do the ImageNet experiments use extracted/pre-trained embedding as latents?**
>
>   A. We follow the experimental setup suggested by VIB (Alemi et al., 2017) for the ImageNet experiments (Section 5.4). To briefly introduce the setup, the 1536-dimensional features are extracted by a pre-trained Inception-ResNet-V2 on ImageNet and then fed into two trainable fully connected layers with 1024 units each, followed by DB or VIB layer which also has 1024 units.

---

### Official Review · AnonReviewer4 · 2020-10-29
**Interesting idea but a little bit too simple?**

**Rating:** 6
**Confidence:** 3

**Review:**

**Summary**
This paper proposes the Drop-Bottleneck (DB) method that performs feature selection during the training with the mutual information. It achieves the state-of-the-art result in few reinforcement learning tasks, and trains a more robust model.

**Originality and significance aspect**
This paper combines mainly two ideas 1) classic feature selection (choose Xis to drop) with respect to the mutual information (between X and Y) 2) Information Bottleneck (IB) formulation that maximizes the prediction-term mutual information term (between Z and Y) and minimizes the compression information (between X and Z) simultaneously. It is actually fairly close to the core idea of the feature selection because it finds the compression by dropping the original feature or not unlike the other IB compression methods. In other words, DB is a modernized version of the feature selection that automatically drops a feature based on the IB-style loss. I would say the idea is not entirely new (somewhat limited), but it could be still useful to the community.

In the significance aspect, I wanted to see authors to apply this method on other noisy setup tasks (e.g. computer vision tasks with noise), outside of the reinforcement learning tasks. The improvement on the RL tasks seems to be substantial; however, I would like to know how DB performs when there is correlation between feature dimensions (see below clarification question to see more details).

**Quality and clarity aspect**

The paper was overall easy to follow. Here are a few questions to authors:

* What if we just drop the feature space only using the mutual information between X and Y and drop them to achieve a similar number of features that was resulted by DB -- it is basically the classic mutual information feature selection. Would that perform as good as DB? Can you make a comparison?  I think this should be one of the baseline. If that's similar to DB, what would be the benefit of DB?
* Does DB always minimize \hat{I}(Z; X) -- with the independent assumption? When some of the Xis are correlated (e.g. consider a vision task), there could be quite a gap between I(Z; X) and the independent assumption version.

**Recommendation**
I think the paper is at the borderline. Looking forward to seeing more discussion with the classic feature selection method and some evaluation on tasks outside of RL (if possible). I would be happy to revisit my score


-----
**Post rebuttal comment**
I thank the authors for the rebuttal. Authors have addressed my concerns and clarified some of the confusing points that I had. I would like to recommend this paper to be accepted.

---

> ### Author Response · Authors · 2020-11-19
> **Author Response to AnonReviewer4**
>
> We appreciate your valuable comments. We will clarify all the concerns in our final draft and make our code public.
>
> 1. **Discussion with the feature selection methods (e.g. empirical comparison with the MI feature selection)**
>
>   A. We agree that the idea to drop a subset of less important feature variables is connected to the feature selection methods. However, one of the important differences of DB from them is that it jointly trains not only the drop probability but also the feature extractor that outputs $X$ (the input to DB) with the prediction term (Section 3.4).
> Also, we updated Section 5.4 of the draft to include the empirical comparison with the mutual information (MI)-based feature selection for dimensionality reduction and adversarial robustness on ImageNet. While DB retains the ImageNet validation accuracy over $71\\%$ with only $4$ feature variables, the feature selection method starts failing to achieve $70\\%$ when there are $<20$ features and its accuracy drops to $9.7\\%$ with $4$ feature variables. Also, the feature selection methods provides almost no adversarial robustness to the targeted $\ell_2$ and $\ell_\infty$ attacks as its resulting attack success rates are all $\geq 97\\%$, whereas DB can reduce the success rate to $20\\%$ for the $\ell_2$ and to $2\\%$ for the $\ell_\infty$ attacks with $8$ features.
>
>
> 2. **Minimization of $\hat{I}(Z; X)$ with the independent assumption**
>
>   A. Since DB encourages dropping of redundant features from $X$, as the training progresses, $Z$ becomes likely to preserve less-correlated features and thus have small total correlation $TC(Z)$, which is the gap between $I(Z; X)$ and $\hat{I}(Z; X)$. That is, although the presented compression term of DB is an upper bound of the true $I(Z; X)$, it is learned to decrease the gap as iteration goes.
>
>
> 3. **Other noisy setup tasks outside of reinforcement learning**
>
>   A. We kindly refer to Appendix B (or C in the updated draft), which experiments the DB’s ability to remove task-irrelevant information on a vision task - classification with Occluded CIFAR dataset.

---

### Official Review · AnonReviewer2 · 2020-10-30
**information bottleneck that drops input features with probability p**

**Rating:** 6
**Confidence:** 3

**Review:**

Summary:
This paper proposes an information bottleneck method, Drop-Bottleneck, that allows the input to be compressed by dropping each input feature with probability p_i. The model then learns the drop probability vector p = [p_1, ... , p_n], where dropping "redundant" features will reduce the "compression penalty" term I(XZ). The approach is demonstrated in experiments in (1) robust exploration setting for RL, (2) adversarial attacks on ImageNet, and (3) an experiment showing that their approach is able to maintain performance on ImageNet with reduced dimensionality.

Evaluation:
I found this paper to be clear and the experiments seem reasonable. I don't know of any prior work that takes this approach. I'm not an RL expert so I won't comment on the strength of the RL results, other than that their methods were clear and they seem to have been careful and fair in choosing baselines. My biggest criticism is that the robustness experiments on ImageNet compare to VIB (Alemi et al 2017) as a baseline, but they should really compare to the more recent adversarial results on Conditional Entropy Bottleneck (another information bottleneck approach that outperforms VIB) given in Fischer and Alemi 2020 (https://arxiv.org/pdf/2002.05380.pdf). I'd like to see that added to the revised version.

---

> ### Author Response · Authors · 2020-11-19
> **Author Response to AnonReviewer2**
>
> We appreciate your valuable comments. We will clarify all the concerns in our final draft and make our code public.
>
> 1. **Comparison with more recent Information Bottleneck methods such as Conditional Entropy Bottleneck (Fischer and Alemi, 2020)**
>
>   A. Following your suggestion, we are implementing and experimenting the CEB method in our settings, but due to the limited time of rebuttal and the fairly large amount of computing resources required by the adversarial tests, we have not finished the experiments yet. Once we obtain results, we will update this reply and the draft again.
>
>
>   **(Updated)** We have uploaded our new revision, which includes the comparison with CEB in the adversarial robustness experiments in Appendix B (due to the 9-page limit). To summarize the results, while CEB also provides the meaningful robustness to the attacks, DB shows the lower (i.e. better) attack success rates in both the targeted $\ell_2$ and $\ell_\infty$ attacks. For the $\ell_2$ attacks, DB achieves $18.5\\%$ at $\beta = 0.01$, and CEB best scores $45\\%$ at (final) $\rho = 3.454$. In case of the $\ell_\infty$ attacks, the attack success rate for DB is $1.5\\%$ and $2.0\\%$ at $\beta = 0.003162$ and $0.01$, and CEB achieves $12.5\\%$ at (final) $\rho = 3.454$.

---

### Author Response · Authors · 2020-11-24
**General Response and Summary of Changes in Revision**

We appreciate all the reviewers for the helpful comments. Thanks to the suggestions, throughout the rebuttal period, we have improved our draft with the following major changes:

1. We added the empirical comparison with the mutual information-based feature selection on the ImageNet tasks. Please refer to our response to AnonReviewer4’s Q1 and the updated results in Section 5.4 for the details.


2. We made a comparison with Conditional Entropy Bottleneck (Fischer and Alemi, 2020) in the adversarial robustness experiments. For its details, please see our updated response to AnonReviewer2's Q1 and Appendix B of the updated revision.


3. We added Grad-CAM (Selvaraju et al., 2017) visualization of task-irrelevant information removal done by DB, in Appendix D of the revised draft.

---

### Decision · Program_Chairs · 2021-01-07
**Final Decision**

**Decision:**

Accept (Poster)

**Comment:**

This paper proposes to enhance the robustness of RL and supervised learning algorithms to noise in the observations by dropping input features that are irrelevant for the task. It relies on the information bottleneck framework (well derived in the paper) and learns a parametric compression of the input features that sets them to zero if they are not relevant for the taskn. The method is extensively evaluated on several RL tasks (exploration in VizDoom and DMLab with a noisy “TV” distractor) and supervised tasks (ImageNet or CIFAR-10 classification with noise).

Reviewers have praised the idea, derivation and writing, as well as the extensive experiments on RL and supervised tasks. Critique focused on:
* the contrived nature of the TV noise (localised always in the same corner of the image -- a standard evaluation according to the authors),
* lack of comparison with other feature selection methods,
* lack of comparison with Conditional Entropy Bottleneck (done during rebuttal),
* more general noise than just specific pixels (clarified by the authors as being the features coming out of a convnet)

Given that the reviewers’ comments were largely addressed by the authors, and given the final scores of the paper, I will recommend acceptance.